# How Do the Outcomes of Radiation-Associated Pelvic and Sacral Bone Sarcomas Compare to Primary Osteosarcomas following Surgical Resection?

**DOI:** 10.3390/cancers14092179

**Published:** 2022-04-27

**Authors:** Alexander L. Lazarides, Zachary D. C. Burke, Manit K. Gundavda, Rostislav Novak, Michelle Ghert, David A. Wilson, Peter S. Rose, Philip Wong, Anthony M. Griffin, Peter C. Ferguson, Jay S. Wunder, Matthew T. Houdek, Kim M. Tsoi

**Affiliations:** 1Division of Orthopaedic Surgery, Department of Surgery, University of Toronto Musculoskeletal Oncology Unit, Sinai Health System, Toronto, ON M5S 1A8, Canada; alexanderleandros.lazarides@sinaihealth.ca (A.L.L.); zachary.burke@sinaihealth.ca (Z.D.C.B.); manitkenan.gundavda@sinaihealth.ca (M.K.G.); rostikn@gmail.com (R.N.); anthony.griffin@sinaihealth.ca (A.M.G.); peter.ferguson@sinaihealth.ca (P.C.F.); jay.wunder@sinaihealth.ca (J.S.W.); 2Division of Orthopaedic Surgery, Department of Surgery, McMaster University, Hamilton, ON L8V 1C3, Canada; ghertm@mcmaster.ca; 3Department of Orthopaedic Surgery, Dalhousie University, Halifax, NS B3H 4R2, Canada; wilsondaj@gmail.com; 4Department of Orthopaedic Surgery, Mayo Clinic, Rochester, MN 55905, USA; rose.peter@mayo.edu (P.S.R.); houdek.matthew@mayo.edu (M.T.H.); 5Department of Radiation Oncology, University of Toronto, Toronto, ON M5T 1P5, Canada; philip.wong@rmp.uhn.ca

**Keywords:** radiation-associated sarcoma, bone sarcoma, osteosarcoma, pelvis, sarcoma, surgical outcomes

## Abstract

**Simple Summary:**

Primary osteosarcomas, spindle cell sarcomas, and radiation-associated sarcomas arising in the pelvis and sacrum (RASB) represent challenging disease processes. The oncologic outcomes are similarly poor between these entities; however, the rates of perioperative death and 5-year disease specific survival for patients with RASB appear to be worse overall. While surgery can result in a favorable outcome for a small subset of patients, surgical treatment is fraught with complications.

**Abstract:**

Radiation-associated sarcoma of the pelvis and/or sacrum (RASB) is a rare but challenging disease process associated with a poor prognosis. We hypothesized that patients with RASB would have worse surgical and oncologic outcomes than patients diagnosed with primary pelvic or sacral bone sarcomas. This was a retrospective, multi-institution, comparative analysis. We reviewed surgically treated patients from multiple tertiary care centers who were diagnosed with a localized RASB. We also identified a comparison group including all patients diagnosed with a primary localized pelvic or sacral osteosarcoma/spindle cell sarcoma of bone (POPS). There were 35 patients with localized RASB and 73 patients with POPS treated with surgical resection. Patients with RASB were older than those with POPS (57 years vs. 38 years, *p* < 0.001). Patients with RASB were less likely to receive chemotherapy (71% for RASB vs. 90% for POPS, *p* = 0.01). Seventeen percent of patients with RASB died in the perioperative period (within 90 days of surgery) as compared to 4% with POPS (*p* = 0.03). Five-year disease-specific survival (DSS) (31% vs. 54% *p* = 0.02) was worse for patients with RASB vs. POPS. There was no difference in 5-year local recurrence free survival (LRFS) or metastasis free survival (MFS). RASB and POPS present challenging disease processes with poor oncologic outcomes. Rates of perioperative mortality and 5-year DSS are worse for RASB when compared to POPS.

## 1. Background

Radiotherapy is frequently utilized as part of curative cancer treatment, either as an adjuvant or with primary curative intent [1]. Despite its widespread use, radiotherapy is associated with short- and long-term toxicity. In the short to mid-term, patients may develop fibrosis, delayed wound healing, and other complications [2,3,4]. In the long term, more serious sequelae may manifest, such as radiation-associated malignancies [1,5,6,7,8,9,10,11,12,13].

Radiation-associated bone sarcomas (RASB) are one such potential secondary malignancy. Prior studies demonstrated that when compared to primary bone sarcomas, perioperative and survival outcomes of RASB are worse [7,8,11]. The pelvis is a common location for RASB due to the frequent use of radiotherapy as part of the treatment for genitourinary, gastrointestinal, and gynecological cancers. Standard treatment for pelvic bone sarcomas includes (neo)adjuvant chemotherapy and surgical resection with or without additional radiotherapy. Surgical options include internal or external hemipelvectomy and, depending on the location and extent of the soft tissue mass, additional vascular, visceral, or soft tissue reconstructive procedures may be required. As such, the decision to pursue curative treatment of a pelvic bone sarcoma requires careful, patient-specific multidisciplinary consideration as the procedures are associated with high morbidity, peri-operative mortality, and long recovery periods [14,15,16]. Patients often require additional surgical procedures to manage post-operative complications, and once discharged from hospital they commonly need extended stays in rehabilitation facilities before being able to return home. Furthermore, patients may be left with chronic pain and/or permanent functional impairments, affecting mobility, bowel, and bladder function, significantly impacting their quality of life [17,18]. As patients with RASB are known to have poor overall survival, the question of whether to offer these patients surgical resection is unclear. Previous small series which examined prognosis of patients showed that patients with RASB frequently present with advanced stages of disease, and even with surgical management survival is poor [8,11].

The purpose of the current study is to examine a patient population from tertiary sarcoma centers in North America that frequently treat RASB to analyze treatment outcomes. We hypothesized that patients with RASB would have worse oncologic and surgical outcomes than patients diagnosed with primary osteosarcoma or spindle cell sarcoma of the pelvis (POPS).

## 2. Methods

This was a retrospective, multi-institution, comparative cohort analysis. Following institutional review board approval at each center, we reviewed all patients diagnosed with a localized radiation-associated pelvic and sacral bone sarcoma between 1 January 1985 and 1 January 2020. Radiation-associated sarcomas were defined as a histologically confirmed bone sarcoma of the pelvis in a previously irradiated field with a minimum 3-year interval between radiation for the primary tumor and the sarcoma diagnosis [7,19,20]. The tumor histology was required to be unique from the original cancer diagnosis. We also identified a comparison group during the same time interval that included all patients diagnosed with a localized, primary pelvic and sacral osteosarcoma or spindle cell sarcoma of bone since they would be eligible for treatment with osteosarcoma-type chemotherapy. We only included patients who underwent surgical resection with curative intent and, for survivors, had at least 12 months follow-up. Exclusion criteria included non-operative treatment, palliative surgical treatment, metastases at diagnosis, and soft tissue sarcomas invading bone.

The primary outcome measure was disease-specific survival. Secondary outcome measures were local recurrence-free and metastasis-free survival as well as post-operative complications. Perioperative death was defined as intraoperative death or death within 90 days of the index surgery.

Grading was classified as grade 2 or grade 3. Spindle cell/pleomorphic sarcoma of bone was graded based on Fédération Nationale des Centres de Lutte Contre Le Cancer (FNCLCC) criteria. Surgery was the mainstay of treatment for both groups. For patients with both a RASB or a POPS, multiagent adjuvant and/or neoadjuvant was considered a part of the standard treatment regimen unless the family refused treatment, or the patient was deemed not to be medically fit for chemotherapy (see below). Radiotherapy was not typically a part of the standard treatment regimen for either patient group, except in select case by case circumstances (see below). Margins were classified based on the R classification as negative (R0), microscopically positive (R1), and grossly positive (R2) [21]. “Good” response to neoadjuvant chemotherapy was considered for necrosis > 90% [22]. Follow up was typically performed on an established schedule for 10 years following surgical resection, with visits every 3 months for the first 2 years, 6 months for the next 3 years until 5 years, and yearly thereafter for a total of 10 years of surveillance. Surveillance consisted of physical exam, pelvic imaging with either a CT or MRI, and a chest CT.

### Statistical Analysis

Disease-specific survival (DSS) was defined as time from surgical resection to confirmed death from disease (RASB or POPS) and included peri-operative mortality (death within 90 days of surgery). Local recurrence-free and metastasis-free survival were defined as time from surgical resection to the first instance of local recurrence or metastasis. RASB and POPS data were compared using unadjusted statistical methods; normally distributed continuous variables were compared using a Student’s *t*-test while nonparametric continuous variables were compared using the Wilcoxon-Mann-Whitney test. A Chi-squared or Fisher’s exact test (for expected counts < 5) were used to compare categorical data. A sensitivity analysis was performed for size and found that 11 cm was a cutoff most associated with DSS, which was used for subsequent survival analyses. The Kaplan Meier method was used to estimate survival and the log rank and Cox proportionate hazards methods were used to compare factors associated with survival. Statistical significance was denoted at a *p* value < 0.05. JMP Pro 15 software (SAS Institute, Inc., Cary, NC, USA) which was used to perform all statistical analyses. 

## 3. Results

### 3.1. Patient Demographics and Treatment Details

We identified 35 patients with RASB of the pelvis and sacrum and 73 patients with POPS that satisfied the inclusion criteria with localized disease of the pelvis or sacrum (Figure 1). Median follow-up from definitive surgery was 95 months (range 20–383 months) for 41 surviving patients. There was no difference between the median follow-up time for survivors between RASB (median 55 months, range 12–325 months) vs. POP (96 months 19–383 months).

In the RASB group, the initial/primary tumor diagnosis was available for 33 patients (Table 1). The most common primary malignancies were genitourinary (*n* = 9 (27%)), bone and soft tissue sarcomas (*n* = 8 (24%)), reproductive (*n* = 7 (21%)), and gastrointestinal (*n* = 6 (18%)). Information on radiation dose and treatment was available for 32 patients. The median radiation dose was 50 Gy (range 25–66 Gy). The median time to secondary malignancy diagnosis was 12.7 years (range 3–27 years). In the POPS group, the histological diagnosis of 67 patients (92%) was conventional osteosarcoma, while the histological subtype of the remaining 6 patients (8%) was primary spindle cell sarcoma of bone. There was a significant difference in histological subtype between groups (*p* = 0.006).

### 3.2. Comparison of Patient and Treatment between RASB and POPS

When comparing the RASB and POPS groups (Table 2), patients with RASB were older (57 years (range 14–84 years) vs. 38 years (range 12–81 years), *p* < 0.001) and were more likely to have isolated disease involving the sacrum (*n* = 15 (43%) vs. *n* = 10 (14%), *p* < 0.001) (Table 2). There was no difference in maximal tumor diameter between patients with a RASB and those with a POPS (median 9 cm (range 3–20 cm) vs. 11 cm (range 3–28 cm), *p* = 0.1).

Patients with a RASB were less likely to receive chemotherapy (*n* = 25 (71%) vs. *n* = 66 (90%), *p* = 0.01). Amongst 87 patients receiving chemotherapy, information on specific regimens was available for 83 patients (95%), including 20 patients with RASB (80%) and 63 with POPS (95%) (Table 3). Patients with RASB were more likely to receive dual agent chemotherapy (*n* = 14 (41%)) while patients with POPS were most commonly treated with three agents (i.e., high-dose methotrexate, Adriamycin, cisplatin; *n* = 38 (54%)) (*p* < 0.001). There was a difference in the median age for dual vs. three agent recipients for patients with a POPS (46 years (range 36–64 years) vs. 29 years (range 13–55 years), *p* < 0.001), but not for patients with a RASB (67 years (range 48–74 years) vs. 65 years (range 23–74 years) vs., *p* = 0.4). For the patients that began chemotherapy, those with RASB were less likely to receive five or more treatment cycles than patients with POPS (*n* = 5 (31%) vs. *n* = 31 (69%), *p* = 0.009). Following neoadjuvant chemotherapy, there was no difference in “good” histological responders between groups. There was no difference in the rates or dose of perioperative radiotherapy administered to each group around the time of surgery for their RASB or POP (Table 3). Of patients receiving radiotherapy, six patients with a RASB (100%) and eight patients with a POPS (89%) received preoperative radiotherapy.

Eighty percent of patients with RASB (*n* = 28) underwent limb salvage compared to 67% with POPS (*n* = 49, *p* = 0.16). When considering combined R1 and R2 margin status, there was no difference in the rate of margin positive surgery (17% for RASB vs. 11% for POPS; *p* = 0.35) between the two groups (Table 2). 

### 3.3. Surgical Outcomes

Patients with a RASB had a higher rate of death within 90 days of surgery (*n* = 6 (17%) vs. *n* = 3 (4%); *p* = 0.03) (Table 2). Of the nine perioperative deaths, reasons for death included two intraoperative deaths, two deaths from post-operative wound infection and sepsis, two deaths from post-operative blood loss, one death from post-operative decline/failure to thrive, one pulmonary embolism, and one death from febrile neutropenia and aplastic anemia following initiation of post-operative chemotherapy. Considering the entire group, the incidence of surgical complications was high, with 85 patients experiencing 101 perioperative complications. However, there was no difference in the incidence of complications between patients with RASB (*n* = 29 (78%)) compared to POPS (*n* = 55 (65%); *p* = 0.28). Wound complications, including flap failures, dehiscence, and deep infections, were the most common post-operative complications following resection of both a RASB (*n* = 15 (51%)) and a POPS (*n* = 33 (60%)). Rates of symptomatic deep vein thrombosis or pulmonary embolus were high, affecting fourteen percent of patients with RASB (*n* = 5) and seven percent with POPS (*n* = 5). 

### 3.4. Oncologic Outcomes

Five-year DSS was worse for patients with RASB compared to those with POPS (31% vs. 54%, *p* = 0.024) (Figure 2a). When controlling for chemotherapy use, however, RASB was no longer associated with worse 5-year DSS (HR 1.4 (0.77–2.55), *p* = 0.27) (Figure 2b). Similarly, when excluding patients who died in the perioperative period (*n* = 9), RASB was no longer associated with worse 5-year DSS (HR 1.6 (0.0.85–2.9), *p* = 0.15). The median time to death for patients with a localized RASB was 11 months (range 0–50 months). Only seven patients with RASB were alive and disease free beyond three years (median 60 months, range 40–320 months). While low numbers precluded statistical analysis, these patients presented with smaller tumors (median 8.5 cm, range 3–16 cm), were younger (median 48 years, range 20–65 years), were more likely to receive chemotherapy (*n* = 6 (86%)), had a “good” histologic response to chemotherapy (*n* = 4 (60%)), and underwent margin negative surgery (*n* = 6 (86%)). 

A univariate model was constructed to identify factors associated with DSS for patients with RASB and POPS who underwent surgical resection (Table 4). For RASB patients, no variables were associated with worse outcomes. For patients with POPS, larger tumor size (HR 2.1 (1–4.49), *p* = 0.01) and lack of chemotherapy (HR 0.4 (0.13–0.92), *p* = 0.03) were associated with worse DSS. 

There was no difference in local recurrence-free survival between patients with RASB and POPS (*p* = 0.5) (Figure 3a). For patients with RASB, seven developed a local recurrence (20%) at a median 5 months (range 3–21.1 months) and none had their recurrent disease resected. Two patients (29%) received chemotherapy and one patient (14%) received radiation therapy for their local recurrence. Three of the seven patients presented simultaneously or subsequently with metastases, and all died of disease. Twenty patients (27.3%) with a POPS developed a local recurrence at a median of 14 months from surgery (range 2–200). Considering the patients who developed a local recurrence, DSS was worse those with RASB compared to POPS (0% vs. 33%, *p* = 0.013) (Figure 3b).

There was no difference in 5-year metastasis free survival for patients with RASB vs. POPS (31% vs. 48% *p* = 0.09) (Figure 4a). Fifteen patients (43%) with RASB developed metastases at a median 8 months (range 3.5–52.1 months), and the majority (75%) occurred within two years of surgery. Four patients (27%) were treated surgically, two patients (13%) received chemotherapy, and one patient (7%) received combined treatment for their metastases. The remainder received no further intervention and were treated palliatively (*n* = 8 (53%)). For patients with a RASB, 5-year DSS was 27% following metastasectomy compared to 12.5% for those treated without surgery. Thirty-three patients with a POPS (45.2%) developed a metastasis at a median of 11 months (range 3–141 months). When comparing patients who developed a metastasis between groups, DSS was poor, with no difference between patients with RASB compared to POPS (19% vs. 14% at 5 years, *p* = 0.67) (Figure 4b).

## 4. Discussion

Radiation-associated sarcomas of the bony pelvis and sacrum are very challenging to treat and are often associated with a poor prognosis. Data to guide decision-making for these patients is limited. Here we present a multi-institutional retrospective review describing the surgical and oncologic outcomes of patients with RASB of the pelvis and sacrum, in comparison with POPS patients. While survival for both groups was found to be poor, 5-year DSS for patients with RASB was worse. The addition of chemotherapy may mitigate this survival difference. Local recurrence or distant metastasis usually occurred within 2 years of surgery. Although surgery offers a chance for a cure in this patient population, post-operative complications are high.

Surgical resection in a previously irradiated field is technically challenging, often necessitating dissection through altered anatomy, fibrosis, and friable neurovascular structures. As such, it has been suggested that the need for amputation to manage these types of secondary malignancies may be higher [7,10,20,23]. However, in this study, the rates of limb salvage did not differ between patients with RASB and POPS, and similar rates of negative margins were achieved. Difficulty in obtaining negative margins has been suggested as one possible reason for poor survival in patients with radiation-associated sarcomas involving bone and soft tissue. A study by Gladdy et al. investigated 130 patients with primary radiation-associated soft tissue sarcoma and found that margin status was an important independent predictor of survival [1], findings that have been recapitulated by others [13]. When considering patients with post-irradiation sarcoma of soft tissue and bone, similar findings have been suggested. Inoue et al. studied 61 patients with radiation-associated bone and soft tissue sarcomas who underwent surgical treatment and found that a wide surgical margin correlated with improved survival [7]. This same relationship has also been suggested when considering only bone sarcomas. Kalra et al. investigated 42 patients with radiation-associated sarcomas of bone [8] and found that complete surgical resection was the only independent determinant for survival. In our patient group, we did not observe an association between margin status and DSS. Interestingly, though, we did find that, amongst patients who developed a local recurrence, DSS was worse for patients with RASB compared to POPS. It has previously been demonstrated that patients who develop a local recurrence following resection of osteosarcoma have worse overall survival [24,25]. The current study appears to reaffirm this relationship, as only 33% of patients with POPS who developed a local recurrence survived. Local relapse after resection of RASB was even worse with no survivors. Taken together, the suggestion from our findings is that local recurrence is a harbinger of worse survival, while margin status may, in and of itself, not be prognostic of survival. However, interpretation of these findings should be cautioned, as only 18% of patients with a RASB and 11% of patients with a POP had a margin positive surgery. As such, our study may be underpowered to detect such an association. 

While negative margin surgical resection is traditionally accepted as the mainstay of curative treatment for patients with pelvic and sacral bone sarcomas, (neo) adjuvant chemotherapy, especially for primary osteosarcoma and spindle cell sarcomas, provides additional disease control. However, the consideration of adjuvant radiotherapy or chemotherapy in the context of RASB is less well understood. Indeed, one of the challenges surrounding RASB is that treatment options are often more limited as patients are older and may have received prior chemotherapy, limiting subsequent options. Shaheen et al. showed that patients with a radiation induced sarcoma of bone of the extremities and pelvis demonstrated improved survival with combined treatment with both chemotherapy and surgery as compared to surgery alone [11]. Bacci et al. similarly found that the use of chemotherapy in RASB resulted in survival rates that were not dissimilar from patients with primary conventional high-grade osteosarcoma [26,27]. However, treatment regimens presented in these studies were heterogenous and these studies did not specifically account for appendicular vs. axial location when investigating the utility of adjuvant therapy. In the present study, patients with RASB were less likely to receive chemotherapy (70.3% received for RASB vs. 90.6% for POPS), though there was no difference in utilization of radiation between the two groups. When considering the individual groups, the addition of chemotherapy was associated with improved survival for patients with POPS (*p* = 0.03) but not RASB (*p* = 0.06). The lack of an association between chemotherapy utilization and improved DSS in RASB likely relates to the small size of that group of patients, as we found no difference in DSS when comparing RASB to POPS when controlling for the use of chemotherapy. In the present study, most patients receiving chemotherapy for a RASB received dual agent chemotherapy regimens that did not include methotrexate and often did not receive a full course of chemotherapy. There was no difference in “good” histological response following neoadjuvant chemotherapy between RASB and POPS. Failure to complete chemotherapy has been suggested as a risk for poorer survival [28] and may be a contributing factor to the poorer DSS observed in the present study. While rates of methotrexate utilization differed, the significance of this finding is less clear. Several randomized trials failed to show a survival benefit with the addition of methotrexate for adult patients compared to dual agent regimens with Adriamycin and cisplatin alone [29,30].

It is important to consider the morbidity and mortality associated with the surgical treatment of RASB when counseling patients. We found that surgery can result in favorable 5-year survival for ~25% of patients; however, it is critical to acknowledge that surgical treatment is fraught with morbidity, with as many as 80% of patients experiencing post-surgical complications and 15% dying in the perioperative period. Local recurrences and metastases occur typically occur within the first year post-operatively, with a median time to local recurrence and metastasis of 5 months and 8 months, respectively. Median survival was only 11 months and disease-free survival beyond 3 years was achieved in only 20% of patients. Given the extensive morbidity and extended recovery required following internal or external hemipelvectomy and sacrectomy [14,15,16], the decision to proceed with surgery requires careful consideration. 

The observation of a survival difference between RASB and POPS is more challenging to elucidate. Interestingly, although patients with a RASB were older and had less intensive chemotherapy treatment, similar oncologic outcomes were achieved overall. When controlling for chemotherapy utilization, DSS was indeed similar between groups, though it is possible that this subanalysis was underpowered to detect such a difference. This points potentially to a comparable tumor biology. In further support of this consideration, margin status was not different between groups and was not associated with DSS. Given that there was no difference in LRFS, then, again, there may not be an explicit difference in the biological aggressiveness of the two groups. The one exception we found was a difference in disease specific survival. This is surprising as there was no difference in LRFS or MFS between groups, so local recurrence and metastasis alone are unlikely to explain this difference. One possible explanation is the large difference in perioperative mortality between groups. Indeed, when excluding patients who died perioperatively, there was no observed difference in DSS between groups, though, again, this subanalysis may be underpowered to detect such a difference. On the other hand, though, it is possible that the high rates of perioperative complications and increased rates of perioperative mortality for RASB patients as compared to POP patients may be related to the challenges of operating after previous surgeries and/or after previous radiation. Thus, while the DSS difference observed may not be explicitly related to tumor biology, it may be explained by the challenges of operating after prior oncologic treatment in the same field. 

There are several important limitations affecting this study. First, given that this is a retrospective review, the study is subject to selection bias. A notable consideration is that this study excluded patients who were treated palliatively. It is reasonable to expect that complications may be higher and outcomes worse in this excluded palliative group and that taken as a whole, the results presented here may be more favorable than if all patients with RASB and POPS of the pelvis and sacrum were included. Furthermore, the exclusion of patients treated without definitive surgery prevents us from understanding the natural history of this disease and weighing the risks and benefits of surgical versus non-operative treatment. Future studies’ investigation RASB of the pelvis and sacrum should consider including patients undergoing palliative treatment. Another limitation is that data regarding functional outcomes was not available for many patients. This was likely because many of the patients did not survive long enough for reasonable functional assessments to be undertaken. As such, we could not draw conclusions as to whether the functional outcomes were different between groups. We did not have data regarding the type of hemipelvectomy or hindquarter amputation performed on each patient. We also acknowledge that there was a higher proportion of sacral tumors in the RASB group, which could also impact decision making regarding surgical candidacy, the extent of surgery and complications. Similarly, there were more patients with spindle cell sarcomas of bone in the RASB group than the POPs group. This may be a contributing factor to the survival difference observed, though there is limited evidence available for this disease entity in the pelvis. Finally, there was incomplete information available for complications managed medically while patients were still an inpatient; as such, it is possible that medically managed complications may have been missed and that the overall rate of complications related to surgery may be higher.

## 5. Conclusions

POPS and RASB involving the pelvis and sacrum present challenging disease processes and their oncologic outcomes are similarly poor. However, the data presented here shows that perioperative and disease specific survival for patients with RASB is even worse than for patients with POPS. While surgery can result in a favorable curative outcome for a small subset of patients, surgical treatment is fraught with complications. As such, careful counseling is necessary to reach a patient-centered decision regarding the suitability and feasibility of proceeding with surgical treatment of these tumors.

## Figures and Tables

**Figure 1 cancers-14-02179-f001:**
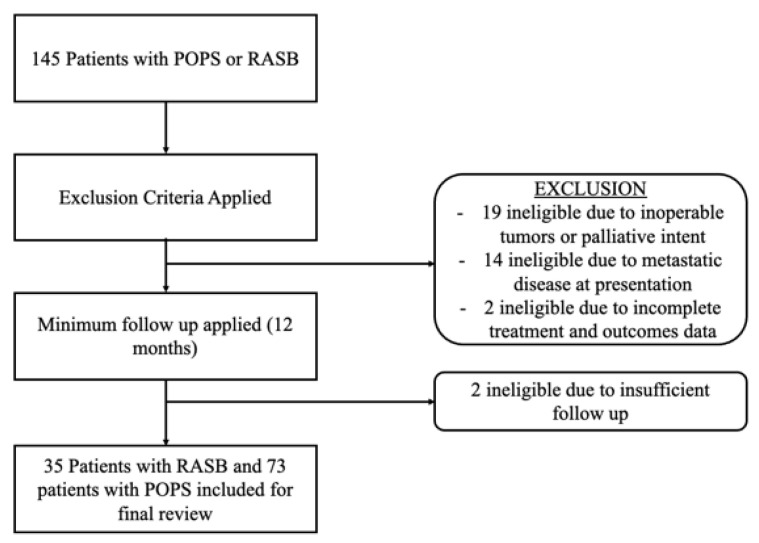
Patient flow chart demonstrating inclusion and exclusion criteria for patients with radiation-associated sarcoma of the pelvis and sacrum and those with primary osteosarcoma or spindle cell sarcoma.

**Figure 2 cancers-14-02179-f002:**
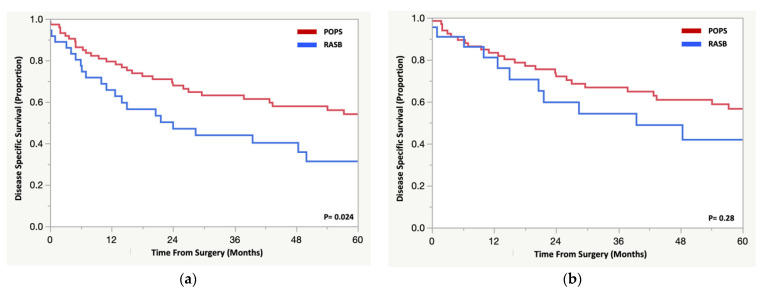
Kaplan Meier analysis for overall disease specific survival between (**a**) patients with a radiation-associated sarcoma of the pelvis/sacrum and a primary osteosarcoma or spindle cell sarcoma. Kaplan Meier analysis for disease specific survival (**b**) controlling for the addition of chemotherapy between patients with a radiation-associated sarcoma of the pelvis/sacrum and a primary osteosarcoma or spindle cell sarcoma.

**Figure 3 cancers-14-02179-f003:**
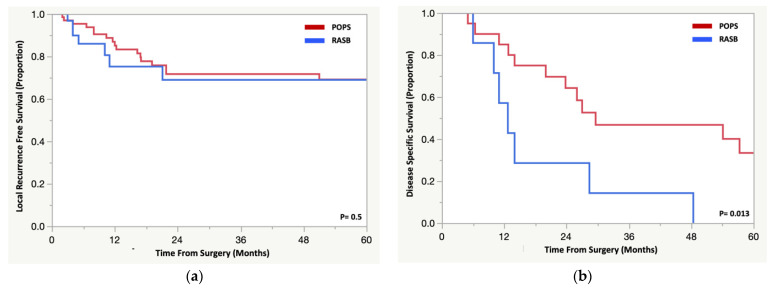
Kaplan Meier analysis comparing patients with a radiation-associated sarcoma of the pelvis/sacrum and a primary osteosarcoma or spindle cell sarcoma for (**a**) local recurrence free survival and (**b**) disease specific survival for patients who developed a local recurrence.

**Figure 4 cancers-14-02179-f004:**
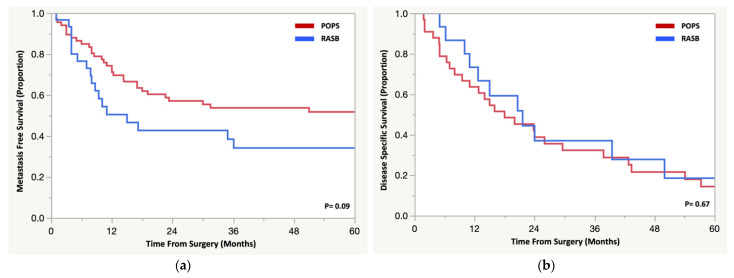
Kaplan Meier analysis comparing patients with a radiation-associated sarcoma of the pelvis/sacrum and a primary osteosarcoma or spindle cell sarcoma for (**a**) metastasis free survival and (**b**) disease specific survival for patients who developed a metastasis.

**Table 1 cancers-14-02179-t001:** Descriptive characteristics of the original cancer diagnosis and treatment, with presenting details on the subsequent histology, for patients with radiation-associated sarcoma of the pelvis and sacrum.

	*n*	%
**Histology (*n* = 33)**		
Gastrointestinal	6	18%
Genitourinary	9	28%
Hematologic (Acute lymphocytic leukemia and unspecified lymphoma)	2	6%
Musculoskeletal/Sarcoma	8	24%
Reproductive	7	21%
Skin	1	3%
**Treatment (*n* = 32)**		
Radiotherapy Alone	6	17%
Surgery with Adjuvant or Neoadjuvant Radiotherapy	29	83%
**Median Dosage (Gy)**	50 (25–66)	
**Median Latency (yrs)**	12.7 (3–27)	
**Secondary Malignancy Histology (*n* = 35)**		
Osteosarcoma	23	66%
Spindle Cell Sarcoma	11	31%
Chondrosarcoma	1	3%

**Table 2 cancers-14-02179-t002:** Patient, tumor, and treatment differences between radiation-associated sarcoma of the pelvis and sacrum and primary osteosarcoma/spindle cell sarcoma.

	Radiation-Associated Sarcoma of Pelvis (*n* = 35)	Primary Osteosarcoma/Spindle Cell Sarcoma of Pelvis (*n* = 73)	
	*n* (%)	*n* (%)	*p*-Value ^#^
**Median Age (years)**	57 (14–84)	38 (12–81)	**<0.001**
**Gender**			
Male	20 (58)	42 (58)	0.94
Female	15 (42)	31 (42)	
**Location**			
Pelvis	19 (54)	50 (68)	**<0.001**
Sacrum	15 (43)	10 (14)	
Combined	1 (3)	13 (18)	
**Median Tumor Size**	9 (3–20)	11 (3–28)	0.1
**Grade**			0.27
Intermediate	4 (11)	4 (6)	
High/undifferentiated	31 (89)	68 (93)	
Unavailable	0	1 (1)	
**Chemotherapy**			**0.011**
Yes	25 (71)	66 (90)	
No	10 (29)	7 (10)	
**Radiotherapy**			0.51
Yes	6 (17)	9 (12)	
No	29 (83)	64 (88)	
**Type of Surgery**			0.16
Limb Salvage	28 (80)	49 (67)	
Amputation	7 (20)	24 (33)	
**Type of Closure**			0.18
Primary	34 (97)	66 (90)	
Flap	1 (3)	7 (10)	
**Margin Status**			0.35
Positive	6 (17)	8 (11)	
Negative	28 (80)	65 (89)	
Unavailable	1 (3)	0	
**Complications**			0.28
Yes	29 (78)	55 (65)	
No	8 (22)	39 (35)	
**Perioperative Death within 90 days of Surgery**			**0.03**
Yes	6 (17)	3 (4)	
No	29 (83)	70 (96)	
**Median time to local recurrence (months)**	5 (3–21.1)	14 (2–200)	0.07
**Median time to metastasis (months)**	8 (3.5–52.1)	11 (3–141)	0.69
**Median time to death (months)**	11 (0–50)	13 (0–57.3)	0.53

# Significance was denoted for *p* < 0.05.

**Table 3 cancers-14-02179-t003:** Adjuvant regimens utilized for patients with localized radiation-associated sarcoma of the pelvis and sacrum and primary localized osteosarcoma/spindle cell sarcoma.

	Radiation-Associated Sarcoma	Primary Osteosarcoma/Spindle Cell Sarcoma	
	*n* (%)	*n* (%)	*p*-Value ^#^
**Number of Agents Received**	34 (97)	70 (96)	**<0.0001**
0 (Family refused or patient not deemed a medical candidate)	14 (41)	7 (10)	
2 (Adriamycin + Cisplatin or Ifosfamide + Adriamycin or Etoposide)	14 (41)	25 (36)	
3 (Methotrexate + Adriamycin + Cisplatin)	6 (18)	38 (54)	
**Chemotherapy Agent Information Unavailable**	1 (3)	3 (4)	
**Number of Chemotherapy Cycles Received**			**0.009**
4 or Fewer Cycles	11 (69)	14 (31)	
5 or More Cycles	5 (31)	31 (69)	
**Chemotherapy Cycle Information Unavailable**	4 (20%)	18 (29%)	
**Percent Necrosis for Patients Receiving Neoadjuvant Chemotherapy**			0.38
>90%	7 (27)	14 (22)	
≤90%	19 (73)	50 (78)	
**Median Radiotherapy Dosing (range, Gy)**	45 (30–50)	52 (15–56.25)	
**Timing of Radiotherapy**			
Neoadjuvant	6 (100)	8 (89)	
Adjuvant	0 (0)	1 (11)	

# Significance was denoted for *p* < 0.05.

**Table 4 cancers-14-02179-t004:** Univariate analysis identifying factors associated with disease specific survival for patients presenting with localized disease.

	Radiation-Associated Sarcoma (*n* = 35)			Primary Osteosarcoma/Spindle Cell Sarcoma (*n* = 73)		
	HR	Confidence Interval	*p*-Value	HR	Confidence Interval	*p*-Value ^#^
**Age (years)**						
<40	ref			ref		
>40	2.6	0.75–8.75	0.13	1.9	0.94–3.84	0.08
**Gender**						
Male	ref			ref		
Female	0.6	0.26–1.53	0.31	0.7	0.32–1.37	0.26
**Location**						
Pelvis	ref					
Sacrum	1.1	0.48–2.59	0.8	1.2	0.41–3.44	0.76
Combined	n/a	n/a	n/a	0.8	0.3–2.08	0.63
**Tumor Size ***						
<11 cm	ref			ref		
>11 cm	2.2	0.88–5.55	0.09	2.1	1–4.49	0.05
**Histologic Grade**						
Intermediate	ref			ref		
High/undifferentiated	0.7	0.21–2.48	0.61	0.9	0.21–3.65	0.64
**Chemotherapy**						
No	ref			ref		
Yes	0.4	0.19–1.02	0.06	0.4	0.13–0.92	**0.03**
**>90% Necrosis following Chemotherapy**						
Yes	-	-	-	ref		
No	-	-	-	3.1	0.72–13.1	0.12
**Radiotherapy**						
No	ref			ref		
Yes	1.7	0.62–4.63	0.3	0.4	0.09–1.58	0.18
**Type of Surgery**						
Amputation	ref			ref		
Limb Salvage	0.7	0.25–1.88	0.46	0.8	0.41–1.73	0.64
**Type of Wound Closure**						
Primary	ref			ref		
Flap	4.4	0.54–35.9	0.17	0.5	0.12–2.13	0.35
**Margin Status**						
Negative	ref			ref		
Positive	1.6	0.6–4.04	0.37	1.6	0.52–4.82	0.41
**Complications**						
No	ref			ref		
Yes	1.9	0.57–6.47	0.3	0.7	0.34–1.43	0.32

***** A sensitivity analysis was performed for size and found that 11 cm as a cutoff most associated with DSS. # Significance was denoted for *p* < 0.05. ref = reference value.

## Data Availability

The data presented in this study are available on request from the corresponding author. The data are not publicly available due to HIPAA and privacy constraints.

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
