# Peer review of "How Do the Outcomes of Radiation-Associated Pelvic and Sacral Bone Sarcomas Compare to Primary Osteosarcomas following Surgical Resection?"

_cancers, 2022, doi:10.3390/cancers14092179_

Round 1

Reviewer 1 Report

The authors present a well-written manuscript comparing radiation- induced and primary bone sarcomas of the pelvic and sacral regions. The topic is interesting and previous literature already dealing these rare situations is very limited. The retrospective design is adequate given the rarity of the disease as well as the used statistical methods. Methods and results are adequately described in general. The conclusions are supported by the data although mentioning rather general observations.

Some comments:

The authors mentioned that cancers after >= 3 years after RT were counted as secondary malignancies. This seems a rather short interval for the development of a secondary cancer after radiation, please comment and add this to the limitations section.

Was median f/u similar in both groups ? If so please state, if not, please discuss and include into the limitations paragraph.

Histologies were not equally distributed between the groups (66% vs 92% osteosarcoma), please discuss and/or include in the limitations paragraph.

Given the different rates of perioperative mortality, it might be fruitful to add a subgroup analysis excluding perioperative deaths in both groups and compare the outcome to get more insights into the biologies of the diseases (see below)

Given the consequences of postop. complications especially deaths for patient counseling, it would be interesting to get more insights into the details of the complications that lead to death (see below) especially as the authors pointed into the direction of a large difference in rather medical complications (like pulmonary embolism) than local complications (like wound complications) in one sentence. Could you add a table with detailed data on the type of complications leading to perioperative mortality in both groups and at least speculate for the reasons in the discussion ?

Discussion:

The discussion includes new and adequate references and deals with some of the major findings in the study. However, I miss a clear discussion of the consequences of the main findings including elaboration of the possible explanations:

Interestingly, LR rate, MFS and median overall survival was not significantly different between the groups, although less patients received additional CHT (and less intensive CHT) and patients were older in the RASB group, while margin status and the use of additional RT was not significantly different. In contrast RASB had significantly worse DSS and increased perioperative mortality. One may argue that although patients were older and had less treatment, similar oncologic outcomes were achieved pointing at a similar biology except for disease specific survival, which seem mainly influenced by perioperative complications/mortality rather than the tumors itself ? Could you comment on this point of view and take this into account during discussion ?

This seems relevant for the following reasons:

With similar RT rates and similar margin status, it seems not surprising (excluding the possibility of a different biological behaviour), that no difference in local recurrence rates could be observed. This might be mentioned in the discussion.

In contrast, with less CHT in the RASB group it seems somewhat surprising that MFS was not statistically different between the groups. However, this might be explained by the low number of patients because a trend (p=0,09) was observed. This might be discussed a little more detailed.

Moreover, it is surprising that there was a clear and significant difference in DSS in the absence of a significant difference of both local recurrences and distant metastases. Although the authors did a secondary analysis and showed that this difference disappeared after controlling for chemotherapy use, it remains difficult to explain. The most reasonable explanation would be to attribute this finding to the large difference in perioperative mortality (promting the question for a secondary analyses excluding the deaths as mentioned above, to get more insight into the biology of the diseases themselves). One may elaborate a little more on these discrepancies in the discussion.

In contrast one may argue that the increased rates of perioperative complications and mortality may have been caused either by the difficulties to perform such large operations after previous surgery and/or after previous radiation (or re-irradiation, as all RASB with RT have been treated neoadjuvantly during the second course of RT). Neoadjuvant RT is known to be associated with increased rates of wound complications per se (even if done as a first course), however the percentage of wound complications was not significantly different. In one sentence, the authors mentioned that the rates of non surgical complications like thrombosis or pulmonary embolus were clearly higher in the RASB group, but did not elaborate further on this issue. However, I feel that this difference might be very important for the interpretation of the data with regard to future treatment decisions and should be reported and discussed more in detail.  

Vielleicht vergleich rechnen ohne die perioperativ gestrobenen für biologisches verhalten ?

Bis discussion gelesen

In this group of patients, we did not observe an association between margin status and DSS.

One may comment that margin status can thus not explain the differences in DSS…

The discussion might be improved if several issues would have been elaborated on more in detail.

Author Response

The authors present a well-written manuscript comparing radiation- induced and primary bone sarcomas of the pelvic and sacral regions. The topic is interesting and previous literature already dealing these rare situations is very limited. The retrospective design is adequate given the rarity of the disease as well as the used statistical methods. Methods and results are adequately described in general. The conclusions are supported by the data although mentioning rather general observations.

Thank you for taking the time to review our manuscript. We appreciate your assistance in improving this work and feel our manuscript is stronger with the help of your comments. Please see specific items below

Some comments:

The authors mentioned that cancers after >= 3 years after RT were counted as secondary malignancies. This seems a rather short interval for the development of a secondary cancer after radiation, please comment and add this to the limitations section.

Thank you for raising this point. We agree that intuitively this may seem a short interval for the development of a secondary cancer after radiation. However, we want to point out that our criteria are rigorous. Radiation-associated sarcomas were defined as a histologically-confirmed bone sarcoma of the pelvis in a previously irradiated field with a minimum 3-year interval between radiation for the primary tumor and the sarcoma diagnosis. The tumor histology was required to be unique from the original cancer diagnosis. This definition has been established previously in the literature and we support this definition with multiple references. Again, we hope to reiterate that this is not an arbitrary definition but one that has been previously established in the literature for considering the diagnosis of a radiation associated sarcoma.

It should be noted that, while 3 years was the minimum cutoff, our median latency was 12.7 years, with a range of 3-27 years.

As this time point has been established with several prior publications, the authors have elected not to include this in the limitations section.

Was median f/u similar in both groups ? If so please state, if not, please discuss and include into the limitations paragraph.

Thank you, we agree that this is an astute point. There was no difference between the median follow- up time for survivors between RASB (median 55 months, range 12-325 months) vs POP (96 months 19-383 months). We have included this in the manuscript.

Histologies were not equally distributed between the groups (66% vs 92% osteosarcoma), please discuss and/or include in the limitations paragraph.

This is an excellent point. We agree that this is an important point to emphasize. We have added additional analysis of this point in the results section and expanded upon this in our limitations

Given the different rates of perioperative mortality, it might be fruitful to add a subgroup analysis excluding perioperative deaths in both groups and compare the outcome to get more insights into the biologies of the diseases (see below)

Given the consequences of postop. complications especially deaths for patient counseling, it would be interesting to get more insights into the details of the complications that lead to death (see below) especially as the authors pointed into the direction of a large difference in rather medical complications (like pulmonary embolism) than local complications (like wound complications) in one sentence. Could you add a table with detailed data on the type of complications leading to perioperative mortality in both groups and at least speculate for the reasons in the discussion ?

The reviewer makes an excellent point. We agree that more specific consideration of the causes of perioperative mortality should be made. We have included more explicit verbiage in the results section outlining the causes of perioperative mortality.

Additionally, we performed a subanalysis excluding perioperative mortality. When excluding patients who died in the perioperative period (n=9), RASB was no longer associated with worse 5-year DSS (HR 1.6 [0.0.85-2.9], p=0.15) It should be noted that this does cut our numbers significantly, so this loss of significance may simply be due to being underpowered

We have expanded the verbiage in the manuscript around the above

Discussion:

The discussion includes new and adequate references and deals with some of the major findings in the study. However, I miss a clear discussion of the consequences of the main findings including elaboration of the possible explanations:

Interestingly, LR rate, MFS and median overall survival was not significantly different between the groups, although less patients received additional CHT (and less intensive CHT) and patients were older in the RASB group, while margin status and the use of additional RT was not significantly different. In contrast RASB had significantly worse DSS and increased perioperative mortality. One may argue that although patients were older and had less treatment, similar oncologic outcomes were achieved pointing at a similar biology except for disease specific survival, which seem mainly influenced by perioperative complications/mortality rather than the tumors itself ? Could you comment on this point of view and take this into account during discussion ?

This seems relevant for the following reasons:

With similar RT rates and similar margin status, it seems not surprising (excluding the possibility of a different biological behaviour), that no difference in local recurrence rates could be observed. This might be mentioned in the discussion.

In contrast, with less CHT in the RASB group it seems somewhat surprising that MFS was not statistically different between the groups. However, this might be explained by the low number of patients because a trend (p=0,09) was observed. This might be discussed a little more detailed.

Moreover, it is surprising that there was a clear and significant difference in DSS in the absence of a significant difference of both local recurrences and distant metastases. Although the authors did a secondary analysis and showed that this difference disappeared after controlling for chemotherapy use, it remains difficult to explain. The most reasonable explanation would be to attribute this finding to the large difference in perioperative mortality (promting the question for a secondary analyses excluding the deaths as mentioned above, to get more insight into the biology of the diseases themselves). One may elaborate a little more on these discrepancies in the discussion.

In contrast one may argue that the increased rates of perioperative complications and mortality may have been caused either by the difficulties to perform such large operations after previous surgery and/or after previous radiation (or re-irradiation, as all RASB with RT have been treated neoadjuvantly during the second course of RT). Neoadjuvant RT is known to be associated with increased rates of wound complications per se (even if done as a first course), however the percentage of wound complications was not significantly different. In one sentence, the authors mentioned that the rates of non surgical complications like thrombosis or pulmonary embolus were clearly higher in the RASB group, but did not elaborate further on this issue. However, I feel that this difference might be very important for the interpretation of the data with regard to future treatment decisions and should be reported and discussed more in detail.   

In this group of patients, we did not observe an association between margin status and DSS.

One may comment that margin status can thus not explain the differences in DSS…

The discussion might be improved if several issues would have been elaborated on more in detail.

The authors thank the reviewer for this astute consideration and insightful analysis of the potential reasons behind such an observed survival difference. We agree that, in its prior form, the manuscript was lacking in an adequate explanation of the circumstances surrounding a survival difference

As such, we have added extensive edits, in addition to further subanalyses, to address the above points.

In the results we add:

“Five-year DSS was worse for patients with RASB compared to those with POPS (31% vs. 54%, p=0.024) (Figure 2A). When controlling for chemotherapy use, however, RASB was no longer associated with worse 5-year DSS (HR 1.4 [0.77-2.55], p=0.27) (Figure 2B).  Similarly, when excluding patients who died in the perioperative period (n=9), RASB was no longer associated with worse 5-year DSS (HR 1.6 [0.0.85-2.9], p=0.15).”

In the discussion we add:

“The observation of a survival difference between RASB and POPS is more chal-lenging to elucidate. Interestingly, although patients with a RASB were older and had less intensive chemotherapy treatment, similar oncologic outcomes were achieved overall. When controlling for chemotherapy utilization, DSS was indeed similar between groups, though it is possible that this subanalysis was underpowered to detect such a difference. This points potentially to a comparable tumor biology. In further support of this con-sideration, margin status was not different between groups and was not associated with DSS. Given that there was no difference in LRFS, then, again, there may not be an explicit difference in the biological aggressiveness of the two groups. The one exception we found was a difference in disease specific survival. This is surprising as there was no difference in LRFS or MFS between groups, so local recurrence and metastasis alone are unlikely to explain this difference. One possible explanation is the large difference in perioperative mortality between groups. Indeed, when excluding patients who died perioperatively, there was no observed difference in DSS between groups, though, again, this subanalysis may be underpowered to detect such a difference. On the other hand, though, it is pos-sible that the high rates of perioperative complications and increased rates of perioper-ative mortality for RASB patients as compared to POP patients may be related to the challenges of operating after previous surgeries and/or after previous radiation. Thus, while the DSS difference observe may not be explicitly related to tumor biology, it may be explained by the challenges of operating after prior oncologic treatment in the same field.”

Reviewer 2 Report

While the authors do not present a novel concept, nor do they use a particularly rigorous scientific technique, the paper is well-designed (albeit retrospective) and very well-written. I do support its publication and do not believe it requires major edits.

1. What is the main question addressed by the research?
Do patients with radiation-associated pelvic sarcomas fare worse than pelvic osteosarcomas?

2. Do you consider the topic original or relevant in the field, and if
so, why?
It is relevant, but not original. This has been studied before. However, of course it is relevant, as prognostication is important in being able to counsel patients and make clinical management decisions as a surgeon.

3. What does it add to the subject area compared with other published
material?
Further corroborates literature already published.

4. What specific improvements could the authors consider regarding the
methodology?
Study is limited by retrospective nature. 

5. Are the conclusions consistent with the evidence and arguments presented and do they address the main question posed?
yes

6. Are the references appropriate?
yes

7. Please include any additional comments on the tables and figures.
n/a

Author Response

Thank you for taking the time to review our manuscript. We appreciate your assistance in improving our manuscript. We agree that this study is inherently limited by its design and retrospective nature. However, this is a topic not easily answered by other means, such as a prospective observational study design, owing to the rarity of radiation associated pelvic and sacral bone sarcomas. To mitigate these limitations, we utilized multiple institutional databases to accrue the largest study sample size to date. We also made a comparison to a relevant but equally rare and challenging disease entity in primary pelvic osteosarcoma. The data presented here is an important contribution to a rare disease entity and we appreciate the reviewer’s consideration.

Reviewer 3 Report

In general:

The subject of this study is the evaluation of treatment results of patients with radiation associated sarcoma of sacrum and pelvis. Further, these results were compared with results of primary sarcoma of this site. This study is designed as multicenter, retrospective analysis.

The manuscript is very detailed, the results are formulated clearly. However, in places it is very difficult to keep the overview because of too many details.

Major issues:

Methods:

In “Results” and “Discussion”, the role of chemotherapy seems to be very important. However, in “Methods”, there are no details to the systemic therapy. In my mind, it is necessary to complete the information.

A part of patients received the radiation as perioperative therapy. I am missing the information regarding the indication for the radiation therapy.

Discussion:

The authors present in results no influence of negative margins and necrosis rate on the DSS. It is discussed in lines 263 and 294 partially. However, what is the interpretation of the author’s results? Please extend the discussion of the points.

Minor issues:

When using an abbreviation for the first time, please give full details:

  • Line 93: FNCLCC
  • Line 34 and 110: DSS

In the text of manuscript the authors use the abbreviation POPS. In all figures the abbreviation POP. Please unify the use of abbreviation.

Line 166: in my opinion, R2 resection contradicts the inclusion criteria of curative therapy.

Line 170-171: The information about the cause of death are very interesting. Please add the information, if available.

Line 177: the citation of figure 2 does not correspond to the content of the figure.

Author Response

In general:

The subject of this study is the evaluation of treatment results of patients with radiation associated sarcoma of sacrum and pelvis. Further, these results were compared with results of primary sarcoma of this site. This study is designed as multicenter, retrospective analysis.

The manuscript is very detailed, the results are formulated clearly. However, in places it is very difficult to keep the overview because of too many details.

 Thank you for taking the time to review our manuscript. We appreciate your assistance in improving our manuscript and feel our manuscript is stronger with the help of your comments. Please see specific items below

Major issues:

Methods:

In “Results” and “Discussion”, the role of chemotherapy seems to be very important. However, in “Methods”, there are no details to the systemic therapy. In my mind, it is necessary to complete the information.

Thank you for raising this point. The authors agree that it is important to include this information and the general treatment strategy for both patient cohorts. We have elaborated upon this in our methodology.

A part of patients received the radiation as perioperative therapy. I am missing the information regarding the indication for the radiation therapy.

Thank you for raising this point. 15 patients total received radiation therapy. The reasons for radiotherapy were variable. Please see the full list below:

  • Three patients treated with curative or palliative intent before orthopaedic oncology surgical intervention
  • One prior intralesional procedure at outside hospital with subsequent rads for local control purposes
  • One giant cell tumor treated preoperatively with rads
  • One patients treated with rads because was thought initially to have metastatic disease from carcinoma
  • One postop rads patients for incomplete margins
  • Specific indication was unavailable for 8 of the patients

Radiotherapy was not generally considered a standard part of the treatment for any of the patients included in this study. We discussed amongst the authors and ultimately felt that explicitly including the above reasons in the manuscript muddied the overall results. We agree that it is important to consider the indications for such treatment, and as such have included verbiage in the methods section indicating that this was not a standard part of the treatment algorithm.

Discussion:

The authors present in results no influence of negative margins and necrosis rate on the DSS. It is discussed in lines 263 and 294 partially. However, what is the interpretation of the author’s results? Please extend the discussion of the points.

Thank you for raising this point. We agree that it is important to expand upon this relationship while also acknowledging the limitations of the present study.

We have added additional verbiage to the discussion section to expand upon this point. Please see the following:

“Difficulty in obtaining negative margins has been suggested as one possible reason for poor survival in patients with radiation associated sarcomas involving bone and soft tissue. A study by Gladdy et al. investigated 130 patients with primary radiation associated soft tissue sarcoma and found that margin status was an important independent predictor of survival,[1] findings that have been recapitulated by others.[13] When considering patients with post irradiation sarcoma of soft tissue and bone, similar findings have been suggested. Inoue et al. studied 61 patients with radiation associated bone and soft tissue sarcomas who underwent surgical treatment and found that a wide surgical margin correlated with improved survival.[7] This same relationship has also been suggested when considering only bone sarcomas. Kalra et al investigated 42 patients with radiation associated sarcomas of bone [8] and found that complete surgical resection was the only independent determinant for survival. In our patient group, we did not observe an association between margin status and DSS. Interestingly, though, we did find that, amongst patients who developed a local recurrence, DSS was worse for patients with RASB compared to POPS. It has been well documented that patients who develop a local recurrence following resection of osteosarcoma have worse overall survival.[24,25] The current study appears to reaffirm this relationship, as only 33% of patients with POPS who developed a local recurrence survived. Local relapse after resection of RASB was even worse with no survivors. It remains unclear if margin status is itself independently associated with survival. The suggestion from our findings is that local recurrence is a harbinger of worse survival, while margin status may, in and of itself, not be prognostic of survival. However, interpretation of these findings should be cautioned, as only 18% (n=6) of patients with a RASB and 11% (n=8) of patients with a POP had a margin positive surgery; as such, our study may be underpowered to detect such an association.”  

Minor issues:

When using an abbreviation for the first time, please give full details:

  • Line 93: FNCLCC
    • Corrected
  • Line 34 and 110: DSS
    • Corrected

In the text of manuscript the authors use the abbreviation POPS. In all figures the abbreviation POP. Please unify the use of abbreviation.

Corrected

Line 166: in my opinion, R2 resection contradicts the inclusion criteria of curative therapy.

This is an astute point and we hope to provide clarity. The authors agree that, when considered in retrospect, an R2 resection would not be considered standard of care for the resection of these bone tumors. However, when we discuss curative intent, we mean to imply the intention at the time of surgery. The goal of surgical treatment at that time was to obtain a margin negative surgery. We explicitly excluded cases in which “tumor debulking” was performed.

In the inclusion criteria was say “We only included patients who underwent surgical resection with curative intent.” Again, this is not meant to imply adequacy of treatment, simply the intent at the time of surgery as an important consideration in this study is: Can these tumors be treated by what we consider standard of care margin negative surgery?

Line 170-171: The information about the cause of death are very interesting. Please add the information, if available.

This is an excellent point. We agree that more detailed information on the nature and cause of perioperative mortality would be beneficial for the readership. We have added greater detail on this in the results section

Line 177: the citation of figure 2 does not correspond to the content of the figure.

Corrected

Reviewer 4 Report

Dear editor,

Thanks for giving this opportunity to review the manuscript entitled “ How do the outcomes of radiation associated pelvic and sacral bone sarcomas compare to primary osteosarcomas following surgical resection?” by  Alexander L. Lazarides , et al. It is an interesting study which attempts to address the main issue of survival and treatment strategy for radiation –associated sarcoma of bony pelvis and sacrum. The specific comments regarding the manuscript are as below:

  1. As we know, radiation-associated sarcoma is one complication of radiation therapy, which is used increasingly as a component of multidisciplinary treatment for many solid tumors. This study demonstrated worse clinical outcomes for patients with RASB than for patients with sporadic histologic counterparts in pelvis and sacrum. Also, it indicated preoperative chemotherapy and negative margin surgical resection are reasonable choices for both RASB and POPS.
  2. The topic is relevent in the field. These original results bring some new insights for strategies in treatment of RASB in pelvic and sacral area.
  3. Radiation-associated sarcomas of the bone and soft tissue can be found in literatures, however, RASB of the bony pelvis and sacrum were foucused on in this study, because they are more challenging to treat in this area and are often associated with a poor prognosis. Moreover, data to guide decision-making for these patients is limited. This original study presented a multi-institutional retrospective review describing the surgical and oncologic outcomes of patients with RASB of the pelvis and sacrum, in comparison with POPS patients.
  4. This is a retrospective review, the study is subject to selection bias as mentioned in the manuscript. Possibly, it may be specific improvement to reveal the unique tumor biology for this challenging disease in the future.
  5. The authors concluded that POPS and RASB involving the pelvis and sacrum present challenging disease processes and their oncologic outcomes are similarly poor. Rates of perioperative mortality and 5-year DSS are even worse for patients with RASB than for patients with POPS. The conclusions consistent with the evidence and arguments presented. The authors fully addressed the main question posed.
  6. The references are appropriate.
  7. There are some number mistakes in the Table 2, I suggest the authors to check and revise.
  8. Abbreviations should be indicated when it first appears. Such as “disease-specific survival”(DSS). 

    Radiation-associated sarcoma is one complication of radiation therapy, which  is used increasingly as a component of multidisciplinary treatment for many solid tumors. This study demonstrated worse clinical outcomes for patients with RASB than for patients with sporadic histologic counterparts in pelvis and sacrum. Also, it indicated preoperative chemotherapy and negative margin surgical resection are reasonable choices for both RASB and POPS. 
    Possibly, it may be interesting to reveal the unique tumor biology for this challenging disease in the future. 

Author Response

Thanks for giving this opportunity to review the manuscript entitled “How do the outcomes of radiation associated pelvic and sacral bone sarcomas compare to primary osteosarcomas following surgical resection?” by  Alexander L. Lazarides , et al. It is an interesting study which attempts to address the main issue of survival and treatment strategy for radiation –associated sarcoma of bony pelvis and sacrum. The specific comments regarding the manuscript are as below:

  1. As we know, radiation-associated sarcoma is one complication of radiation therapy, which is used increasingly as a component of multidisciplinary treatment for many solid tumors. This study demonstrated worse clinical outcomes for patients with RASB than for patients with sporadic histologic counterparts in pelvis and sacrum. Also, it indicated preoperative chemotherapy and negative margin surgical resection are reasonable choices for both RASB and POPS.

Thank you for taking the time to review our manuscript. We appreciate your assistance in improving this work and feel our manuscript is stronger with the help of your comments. Please see specific items below

  1. The topic is relevant in the field. These original results bring some new insights for strategies in treatment of RASB in pelvic and sacral area.

Thank you

  1. Radiation-associated sarcomas of the bone and soft tissue can be found in literatures, however, RASB of the bony pelvis and sacrum were foucused on in this study, because they are more challenging to treat in this area and are often associated with a poor prognosis. Moreover, data to guide decision-making for these patients is limited. This original study presented a multi-institutional retrospective review describing the surgical and oncologic outcomes of patients with RASB of the pelvis and sacrum, in comparison with POPS patients.

Thank you

  1. This is a retrospective review, the study is subject to selection bias as mentioned in the manuscript. Possibly, it may be specific improvement to reveal the unique tumor biology for this challenging disease in the future.

Thank you for your insights, we agree and have included this as a component of our limitations.

  1. The authors concluded that POPS and RASB involving the pelvis and sacrum present challenging disease processes and their oncologic outcomes are similarly poor. Rates of perioperative mortality and 5-year DSS are even worse for patients with RASB than for patients with POPS. The conclusions consistent with the evidence and arguments presented. The authors fully addressed the main question posed.

  1. The references are appropriate.

Thank you

  1. There are some number mistakes in the Table 2, I suggest the authors to check and revise.

Thank you for your astute reading of the manuscript. For some of the variables, such as grade and margin status, data was not available for some of the patients. We have added a row indicating missing data if applicable

  1. Abbreviations should be indicated when it first appears. Such as “disease-specific survival”(DSS). 

Done

Radiation-associated sarcoma is one complication of radiation therapy, which  is used increasingly as a component of multidisciplinary treatment for many solid tumors. This study demonstrated worse clinical outcomes for patients with RASB than for patients with sporadic histologic counterparts in pelvis and sacrum. Also, it indicated preoperative chemotherapy and negative margin surgical resection are reasonable choices for both RASB and POPS. 

Possibly, it may be interesting to reveal the unique tumor biology for this challenging disease in the future. 

Thank you again for your astute reading of the manuscript